# The Method of Segmenting the Early Warning Thresholds Based on Fisher Optimal Segmentation

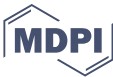

**Xiangyu Li [1], Tianjie Lei [2,\*], Jing Qin [1,\*], Jiabao Wang [3], Weiwei Wang [4,†], Baoyin Liu [5], Dongpan Chen [6,†], Guansheng Qian [2,†], Li Zhang [7,†] and Jingxuan Lu [1,†]**

[1] State Key Laboratory of Simulation and Regulation of Water Cycle in River Basin, China Institute of Water Resources and Hydropower Research, Beijing 100038, China

[2] Institute of Environment and Sustainable Development in Agriculture, Chinese Academy of Agricultural Sciences, Beijing 100081, China

[3] College of Geoscience and Surveying Engineering, China University of Mining and Technology, Beijing (CUMTB), Beijing 100083, China

[4] China Electronic Greatwall ShengFeiFan Information System Co., Ltd., Beijing 102200, China

[5] Institutes of Science and Development, University of Chinese Academy of Sciences, Beijing 100190, China

[6] Faculty of Information Technology, Beijing University of Technology, Beijing 100124, China

[7] Beijing Institute of Technology, Beijing 100081, China

[\*] Correspondence: leitianjie@caas.cn (T.L.); qinjing@iwhr.com (J.Q.); Tel.: +86-108-210-5990 (T.L.)

[†] Co-firstauthors: These authors contributed equally to this work.

**Abstract:** Most slope collapse accidents are indicated by certain signs before their occurrence, and unnecessary losses can be avoided by predicting slope deformation. However, the early warning signs of slope deformation are often misjudged. It is necessary to establish a method to determine the appropriate early warning signs in sliding thresholds. Here, to better understand the impact of different scales on the early warning signs of sliding thresholds, we used the Fisher optimal segmentation method to establish the early warning signs of a sliding threshold model based on deformation speed and deformation acceleration at different spatial scales. Our results indicated that the accuracy of the early warning signs of sliding thresholds at the surface scale was the highest. Among them, the early warning thresholds of the blue, yellow, orange, and red level on a small scale were 369.31 mm, 428.96 mm, 448.41 mm, and 923.7 mm, respectively. The evaluation accuracy of disaster non-occurrence and occurrence was 93.25% and 92.41%, respectively. The early warning thresholds of the blue, yellow, orange, and red level on a large scale were 980.11 mm, 1038.16 mm, 2164.63 mm, and 9492.75 mm, respectively. The evaluation accuracy of disaster non-occurrence and occurrence was 97.22% and 97.44%, respectively. Therefore, it is necessary to choose deformation at the surface scale with a large scale as the sliding threshold. Our results effectively solve the problem of misjudgment of the early warning signs of slope collapse, which is of great significance for ensuring the safe operation of water conservation projects and improving the slope deformation warning capability.

**Keywords:** fisher optimal segmentation method; warning; threshold determination; regression model





## 1. Introduction

Significant slope collapse is one of the most destructive and costly natural disasters that affects the safety and development of water engineering. This phenomenon is exacerbated by the over-exploitation of natural resources [1–5]. For the safety of water engineering, water resources, ecology, and society, it is important to establish an early warning model [6–10]. This problem has evoked high levels of interest outside of the scientific community. However, methods for establishing early warning signs are not always perfect, and misjudgment of the early warning signs often occurs [11–13].

Correctly determining the early warning signs in sliding thresholds is the basis for establishing an early warning system. There have been many previous studies focusing

on the determination of the early warning signs in sliding thresholds in the field of slope collapse disasters. For example, He et al. established a slope monitoring and prediction system for landslide displacements; the evaluation model of the sensitivity of steep slopes was established by combining an AHP model and expert opinions, and the classification standard and threshold of the steep slope were determined [14]. B Ahmed et al. quantitatively analyzed the correlation between the precipitation threshold and the different risk areas, to judge the influence of precipitation threshold on landslide displacement in different risk areas, and determined the landslide warning threshold in different risk areas [15]. At present, the comparison method, the fluctuation method, the critical value method, the comprehensive evaluation method, the expert consultation method, and other methods are among the most common methods used to determine the early warnings signs in sliding thresholds [16]. Among them, the comparison method includes the median principle method, average method, and majority principle method. The method has been used to determine the early warnings signs in sliding thresholds by calculating the medians or the averages [17]. However, the method is easily affected by the averages of the indicators. Meanwhile, the fluctuation method includes the parameter principle method, correlation principle method, and fluctuation principle method. This method has been used to determine the early warnings signs in sliding thresholds by analyzing the fluctuations in historical data according to the averages and the standard deviation [18]. However, this method was affected by historical data and the accuracy of the early warning signs in sliding thresholds is low when calculating them using the historical data. The expert consultation method is a common method for determining the early warnings signs in sliding thresholds, which can reflect the risk preference of decision makers [19]. However, this method is affected by human factors. The critical value method has stronger pertinence and accuracy and has more choices in the process of determining the early warning threshold. It is more flexible to apply and is suitable for the analysis of specific events. It is necessary to rely on methods such as model analysis and risk evaluation for the setting of early warning thresholds [20]. However, this method is mainly applicable to the analysis of specific events. The comprehensive evaluation method has fewer requirements for quantitative data. Through mathematical analysis and processing of the relationship between multiple variables, it is possible to obtain a quantitative value that is closer to the actual situation, so this method should be applied to study areas with many variables or difficult quantitative factors [21]. However, the mathematical calculation in this method is relatively complicated, and there is a certain subjectivity for the weights between multiple variables.

In view of this, on the basis of not changing the time series used, the Fisher optimal segmentation method can not only comprehensively consider multi-factor indicators but also scientifically show the influence of the change in the number of stages on the calculation results to determine the optimal number of stages and shows a strong advantage. This method can solve the interference of human factors and subjectivity. At present, this method has been widely applied in the fields of agriculture, meteorology, and geological disaster forecasting [22–24]. However, it has not been widely applied in the process of determining the early warning thresholds for steep slope deformation. Here, we applied the Fisher optimal segmentation method to the field of determining the early warning thresholds for steep slope deformation for the first time. Both the deformation speed and acceleration, which affect the slope deformation at different spatial scales, were selected as indicators. The sample data were then scientifically segmented by the Fisher optimal segmentation model to identify the early warning thresholds. Finally, af regression model was used to evaluate the accuracy of the early warning threshold model. The technical route of this method is shown in Figure 1. This solves the limitation of prediction, which can be interfered with by misjudgment of the early warning signs. Our proposed method can improve the early warning capabilities and avoid the occurrence of misjudgment, providing a simple and efficient new method for the early warning signs of slope collapse disasters.

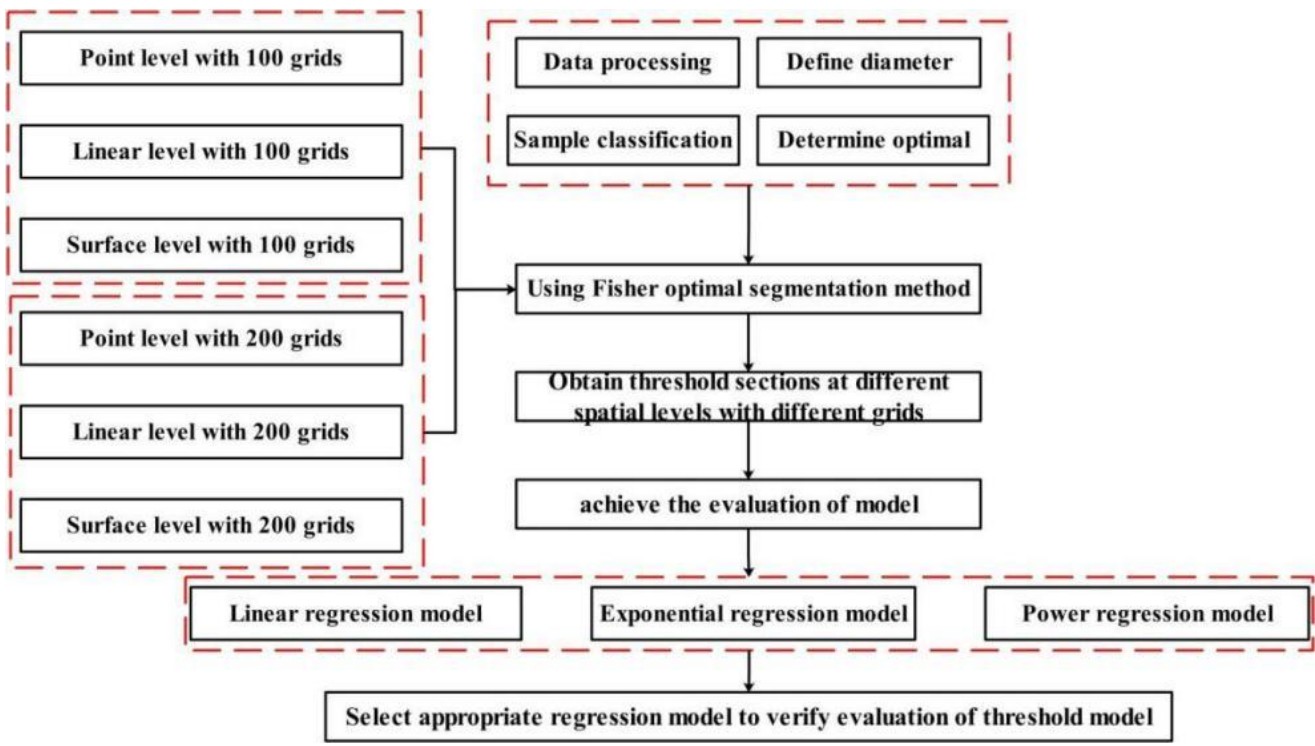

**Figure 1.** The technical route of the early warning threshold model.

## 2. Materials and Methods

### 2.1. Study Area

The study region is a steep slope of the Hongshiyan water control project; it is located at the junction of Lijiashan village, Huodehong Township, Ludian County and Hongshiyan village, Baogunao Township, Qiaojia County. An earthquake on August 3, 2014 caused many geological disasters such as landslides and collapses, forming a barrier lake. Among these disasters, the super large-scale collapse of the right slope and the collapse of the front edge of the left slope formed a large landslide dam with a height of 600 m and a volume of about 12 million cubic meters. The location of the study area is shown in Figure 2. After the earthquake, the landslide dam was extremely unstable, and secondary disasters are very serious. Meanwhile, there are often rocks falling off in the local areas, which can easily cause damage to construction equipment and threaten the personal safety of construction personnel in the process of construction. Therefore, quantitatively establishing the early warning signs of sliding thresholds is important to protect the safety of construction.

### 2.2. Data Source

In this study, we needed to obtain the deformation data and the DEM data. The DEM data were derived from UAV remote sensing technology; the slope orthophoto image model was constructed by UAV remote sensing data and geographic data, and their accuracy was on the centimeter~decimeter level. Meanwhile, the data of slope deformation from 1st January 2019 to 31st December 2019 were obtained from ground-based synthetic aperture radar. The data mainly included the deformation, deformation speed, and deformation acceleration. We obtained the deformation speed and deformation acceleration at different spatial scales by pretreating deformation data using the method of mathematical statistics. Ground-based synthetic aperture radar is suitable for monitoring the slow deformation stage of landslides, so our proposed method focused on analyzing the early warning signs of sliding thresholds at the slow deformation stage of landslides.

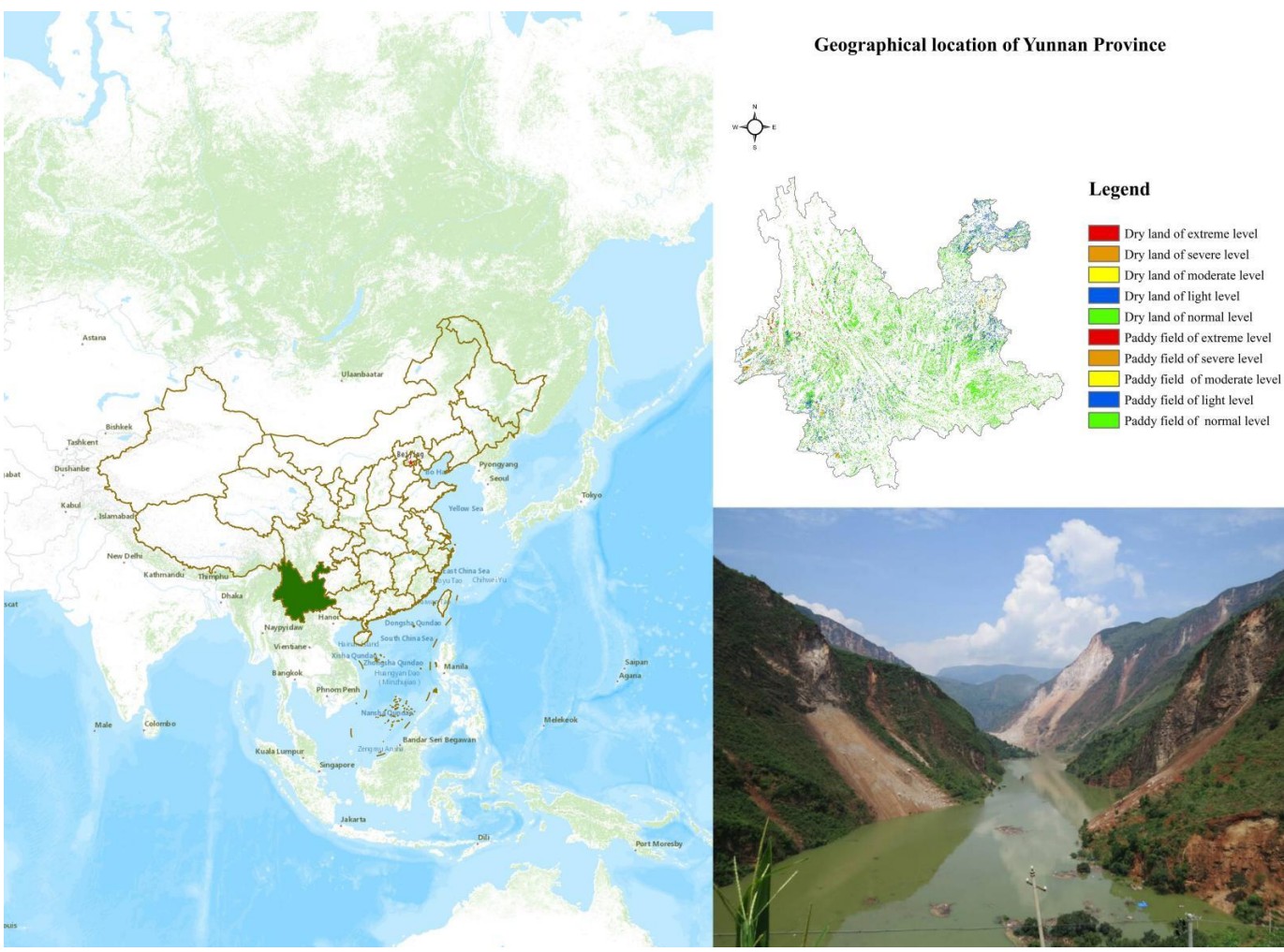

**Figure 2.** The location of the slope in the Hongshiyan water project.

*2.3. Landslide Type*

Landslides form in different geological environments and exhibit various forms and characteristics. The purpose of landslide classification is to summarize the various environmental and phenomenon characteristics of landslide action with various factors, and correctly distinguish some rules on landslide action. In practice, scientific landslide classification can be used to guide exploration work, identify the possibility of landslides, and make the corresponding measures.

According to the classification of the landslide speed, when a landslide belongs to the type of slow deformation, the daily slope deformation is several centimeters to tens of centimeters. A large range of landslide phenomena can only be found through monitoring, and the local range of landslide phenomena is more obvious. Meanwhile, landslides are mainly caused by excavation of the mountain, which leads to the deformation of the upper rock layer sliding and squeezing the lower part. A sliding landslide is characterized by a higher sliding speed. The slope is an engineering landslide between 10 degrees and 45 degrees with a steep downwards slope and a gentle upwards slope. The thickness of the landslide was about 10 m, and the volume of the landslide was less than 1 million cubic meters [25–28].

Therefore, our study area belongs to the type of slow landslide, engineering landslide, minor landslide, and sliding landslide, and our proposed method should be suitable for these types.

### 2.4. Fisher Optimal Segmentation Method

The Fisher optimal segmentation method is a method of clustering and partitioning ordered sample sequences, the principle of this method is mainly segmenting the ordered sample sequence, and the square deviation of each segmentation should be minimized so that the difference within the segmentation is the smallest and the difference between the segmentation is the largest [29–33].

Step 1: First of all, $\{X_1, X_2, X_3, \ldots, X_n\}$ was set as the ordinal sample sequence, then each sample was contained in m evaluation indicators, and matrix X was built. The matrix X is shown in Formula (1).

$$X_{ij} = \begin{bmatrix} x_{11} & x_{12} & \cdots & x_{1m} \\ x_{21} & x_{22} & \cdots & x_{2m} \\ \cdots & \cdots & \cdots & \cdots \\ x_{n1} & x_{n2} & \cdots & x_{nm} \end{bmatrix} \tag{1}$$

The meaning, values, and units of each indicator were different, so matrix X should be normalized. The normalization process is shown in Formula (2).

$$x'_{ij} = \left(x_{ij} - x_{min.j}\right) / \left(x_{max.j} - x_{min.j}\right) \tag{2}$$

where $x'_{ij}$ indicates a normalized matrix; $x_{min.j}$ indicates the minimum of the jth indicator; and $x_{max.j}$ indicates the maximum of the jth indicator.

Step 2: First of all, $\{x_s, x_{s+1}, x_{s+2}, \ldots, x_t\}$. was set a certain sequence, and t > s. At the same time, $R(i, j)$ was set as the diameter of class. The diameter of class is shown in Formula (3).

$$R(i, j) = \sum_{r=i}^{j} (x_r - \overline{x_{st}})^2 \tag{3}$$

where $x_r$ indicates the eigenvalues in the sequence sample; $\overline{x_{st}}$ indicates the average of all data in the series; and $\overline{x_{st}}$ is shown in Formula (4).

$$\overline{x_{st}} = \frac{1}{t - s + 1} \sum_{k=i}^{j} x_k \tag{4}$$

Step 3: First of all, n samples were divided into k categories, and the k categories were $P_1 = \{x_{s_1}, x_{s_1+1}, \ldots, x_{s_2-1}\}, P_2 = \{x_{s_2}, x_{s_2+1}, \ldots, x_{s_3-1}\}, \ldots,$ and $P_k = \{x_{s_k}, x_{s_k+1}, \ldots, x_{s_{k+1}-1}\}$. The objective function result is shown in Formula (5).

$$F(n, k) = \min \sum_{j=1}^{k} R(s_j, s_{j+1} - 1) \tag{5}$$

Step 4: When the values of n and k were determined, the sum of the class diameters could be calculated. When the sum of the class diameters was minimized, the obtained solution was the optimal solution. The result of the Fisher optimal segmentation method is shown in Formulas (6) and (7).

$$F(n, 2) = \min_{2 \le s \le n} [R(1, s - 1) + R(s, n)] \tag{6}$$

$$F(n, k) = \min_{k \le s \le n} [F(s - 1, k - 1) + R(s, n)] \tag{7}$$

where $s_k$ indicates the dividing point when finding the minimum of the optimal solution. At this time, the optimal solution is shown as $F(n, k) = F(s_k - 1, k - 1) + D(s_k, n)$; $\{x_{s_k}, x_{s_k+1}, \ldots x_n\}$ is the kth category; and $s_{k-1}$ indicates the dividing point. The optimal solution with the dividing point $s_{k-1}$ is shown as $F(s_{k-1}, k - 1) = F(s_{k-1} - 1, k - 2) +$

$D(s_{k-1}, n)$, so $\{x_{s_{k-1}}, x_{s_{k-1}+1}, \ldots x_n,\}$ is the (k-1)th category. Finally, all the dividing points and optimal solutions were calculated according to the above methods.

Step 5: A graph showing that the objective function $F(n, k)$ varies with the category k was completed according to the classification results, and the graph was shown as $F(n, k)$-k. k indicates the classification of the objective function of finding an obvious turning point in the graph. When the classification is equal to k, the loss function was the largest, so k was the optimal segments.

### 2.5. Accuracy Evaluation

On the basis of the construction of the early warning threshold model, it is necessary to further evaluate the accuracy of the model. The linear regression model is a mathematical model that reflects the correlations between the dependent variables and the independent variables. The model contains independent variables and dependent variables, and a straight line is used to approximately express the relationships between the dependent variables and the independent variables [34–38]. Then, the model is a single-element regression model and is shown in Formula (8).

$$Y = \beta_1 + \beta_2 X + \varepsilon \tag{8}$$

where $\beta_1$ indicates the intercept of function Y, $\beta_2$ indicates the slope, X indicates the independent variables, and $\varepsilon$ indicates the random error.

The coefficient of determination indicates the part explaining the dependent variables according to the changes in the independent variables in the linear regression model, and so the coefficient of determination judges the degree of fit of the model. At the same time, the range of the coefficient of determination is between zero and one, and when the value is closer to one, it indicates that the degree of fit of the model is better. Conversely, when the value is closer to zero, it indicates that the degree of fit of the model is worse. The coefficient of determination is shown in Formula (9).

$$R^2 = \frac{SSR}{SST} = 1 - \frac{SSE}{SST} \tag{9}$$

where SST indicates the sum of square deviation. It is the sum of the square between the average dependent variables and the actual dependent variables, and it indicates the overall fluctuation in the values of dependent variables. SSR is the regression sum of squares. It is the sum of the square between the average dependent variables and the regression dependent variables, and it indicates the changes in the sum of deviation about the linear relation between the dependent variables and the independent variables.

In order to evaluate the accuracy of the early warning threshold model, the regression model was used. Then, the linear regression model, power regression model, and exponential regression model could be used to perform regression analysis at different sliding threshold intervals. At the same time, the regression model with a high fit was selected to evaluate the accuracy of the early warning of sliding thresholds at different sliding threshold intervals by comparing the coefficient of determination of three regression models.

## 3. Results

### 3.1. Indicators Selection

In this study, a ground-based synthetic aperture radar was used to collect the data of slope deformation, such as speed and acceleration, in 2019. Ground-based synthetic aperture radars are high-resolution radars characterized by a high-resolution ratio and large scale, enabling real-time automatic all-day and all-weather monitoring, and the radar should obtain high-resolution radar images under extreme environmental conditions. The data of each grid can be obtained by data processing, and the accuracy can reach sub-millimeter accuracy. Therefore, we should select the deformation speed and deformation acceleration at different spatial scales as the evaluation indicators under a different number

of grids, and the evaluation indicators need to be counted and redefined in the process of establishing the early warning threshold model. The meanings of evaluation indicators are shown in Table 1.

**Table 1.** The meanings of all indicators.

| Number | Indicators | Meanings |
| --- | --- | --- |
| Number one | Point speed | The maximum speed under all grids in the slope DEM model |
| Number two | Linear speed | The sum of speeds in the grids passed by terrain lines in the slope DEM model |
| Number three | Surface speed | The sum of deformation speeds along the same direction under all grids in the slope DEM model |
| Number four | Point acceleration | The maximum acceleration under all grids in the slope DEM model |
| Number five | Linear acceleration | The sum of accelerations in the grids passed by terrain lines in the slope DEM model |
| Number six | Surface acceleration | The sum of deformation accelerations along the same direction under all grids in the slope DEM model |

### 3.2. The Early Warning Signs of Sliding Thresholds Result

The statistical point speed, point acceleration, linear point, linear acceleration, surface speed, and surface acceleration deformation indicators were recorded from 1st January 2019 to 31st December 2019. Each indicator and meaning were different, so it was necessary to normalize indicators. In this study, the DEM model was divided into 100 grids and 200 grids, the data were counted every day, and then the early warning signs of sliding thresholds were determined according to the principle of the Fisher optimal segmentation. The results of the Fisher optimal segmentation model are shown in Figure 3.

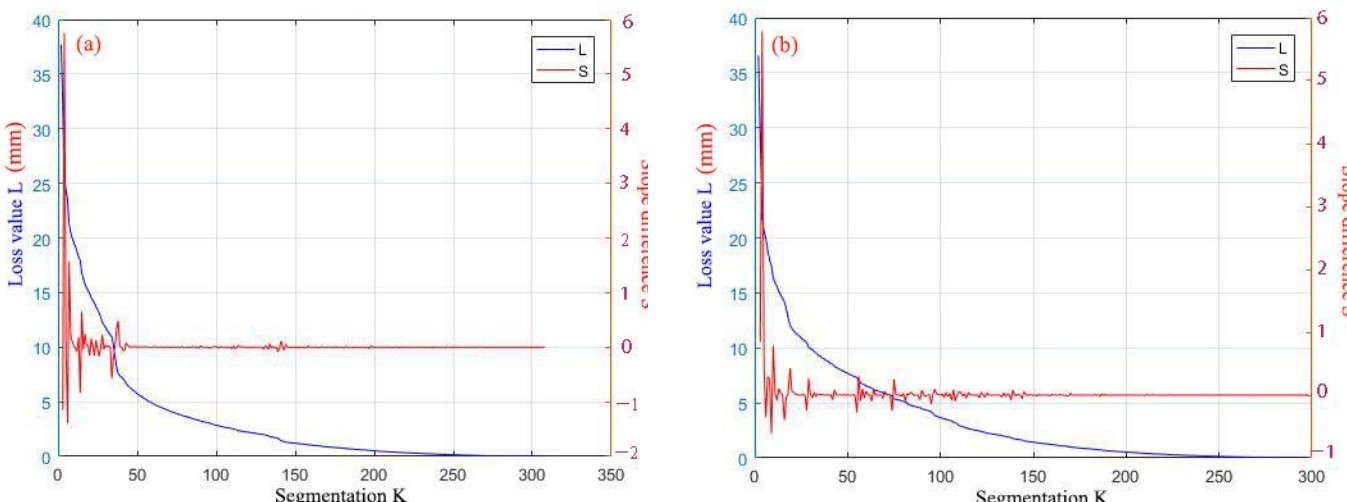

**Figure 3.** The results of Fisher optimal segmentation method under the different scales. (**a**) 100 grids; (**b**) 200 grids.

Figure 3 shows that the results from different numbers of grids have clear inflection points when k is four, and the slope difference reaches a peak; therefore, the optimal number of sample divisions should be four. The slope difference presented in red reached its peak. The loss value presented in blue is the value corresponding to the slope difference. This could be explained by the fact that under the 100 grids and 200 grids models, there were four levels such as blue warning, yellow warning, orange warning, and red warning. The results of the early warning signs of sliding thresholds are shown in Tables 2 and 3.

**Table 2.** The results of deformation threshold intervals with a small scale (mm).

| Early Warning Level | Point Scale | Linear Scale | Surface Scale |
|---|---|---|---|
| Non-occurrence | (0, 59.2) | (0.122.29) | (0.369.31) |
| Blue warning | (59.2, 67.78) | (122.29, 159.45) | (369.31, 428.96) |
| Yellow warning | (67.78, 84.66) | (159.45, 169.28) | (428.96, 448.41) |
| Orange warning | (84.66, 85.84) | (169.28, 220.93) | (448.41, 923.7) |
| Red warning | (85.44, +∞) | (220.93, +∞) | (923.7, +∞) |

**Table 3.** The results of deformation threshold intervals with large scale (mm).

| Early Warning Level | Point Scale | Linear Scale | Surface Scale |
|---|---|---|---|
| Non-occurrence | (0.137.95) | (0.214.04) | (0.980.11) |
| Blue warning | (137.95, 153.82) | (214.04, 224.90) | (980.11, 1038.16) |
| Yellow warning | (153.82, 200.35) | (224.90, 235.44) | (1038.16, 2164.63) |
| Orange warning | (200.35, 769.14) | (235.44, 587.86) | (2164.63, 9492.75) |
| Red warning | (769.14, +∞) | (587.86, +∞) | (9492.75, +∞) |

### 3.3. Accuracy Evaluation

We took the sliding threshold intervals on the 100-grid scale as an example. The deformation data were segmented by using the Fisher optimal segmentation method, and then the linear regression model, power regression model, and exponential regression model could be used for each threshold interval to perform regression analysis. The regression analysis results of all sliding threshold intervals are shown as follows.

Figure 4 shows the results of regression analysis in the first threshold interval with different regression models. The results of the regression relationship are shown in Formulas (10)–(12).

$$y = 0.6097x + 20.597 \tag{10}$$

$$y = 19.91e^{0.015x} \tag{11}$$

$$y = 3.6864x^{0.6814} \tag{12}$$

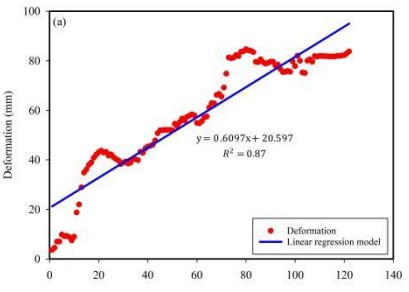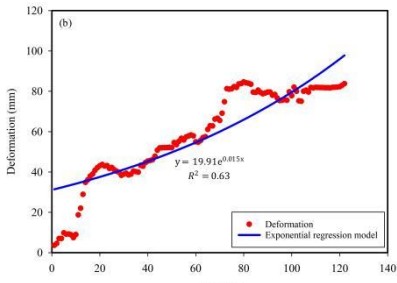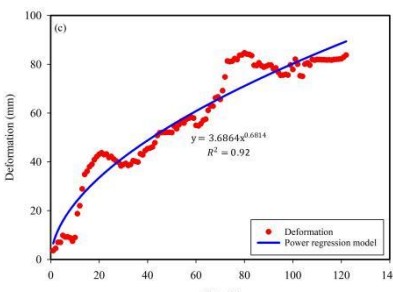

**Figure 4.** The results of regression analysis in the first threshold interval with different regression models. (**a**) Linear regression model; (**b**) exponential regression model; (**c**) power regression model.

The coefficients of determination of the linear regression model, exponential regression model, and power regression model were 0.87, 0.63, and 0.92, respectively. Therefore, the regression model in the first threshold interval was similar to a power regression model.

Figure 5 shows the results of regression analysis in the second threshold interval with different regression models. The results of the regression relationship are shown in Formulas (13)–(15).

$$y = -0.0762x + 80.96 \tag{13}$$

$$y = 80.947e^{-0.001x} \tag{14}$$

$$y = 85.1x^{-0.026} \tag{15}$$

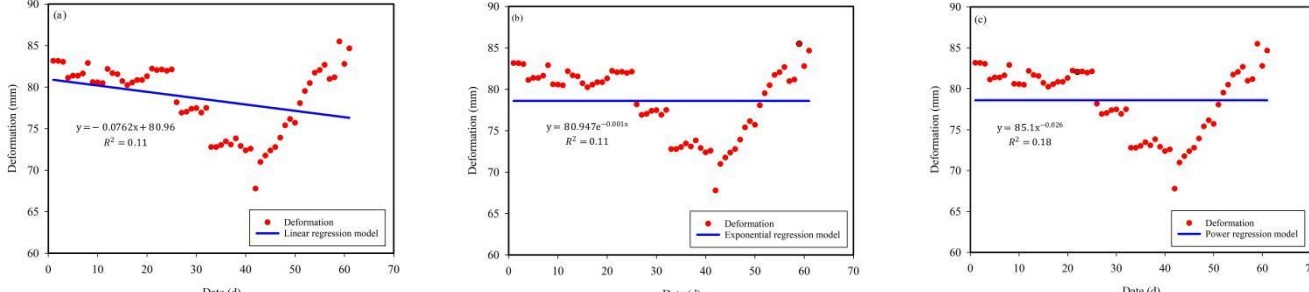

**Figure 5.** The results of regression analysis in the second threshold interval with different regression models. (**a**) Linear regression model; (**b**) exponential regression model; (**c**) power regression model.

However, the range of data fluctuation in the second threshold interval was large and may impact the fit of the regression model. Overall, the coefficients of determination of the linear regression model, exponential regression model, and power regression model were low, at 0.11, 0.11, and 0.18, respectively. Therefore, the regression model in the second threshold interval was similar to a power regression model.

Figure 6 shows the results of regression analysis in the third threshold interval with different kinds of regression models. The results of the regression relationship are shown in Formulas (16)–(18).

$$y = 25.56x + 3.9253 \tag{16}$$

$$y = 86.79e^{0.0894x} \tag{17}$$

$$y = 46.313x^{0.7719} \tag{18}$$

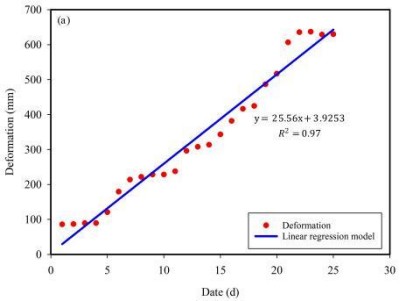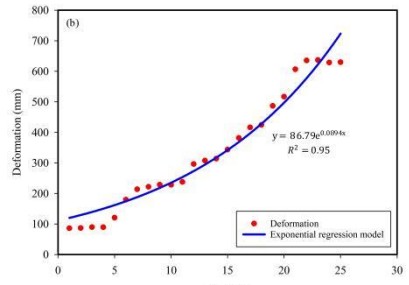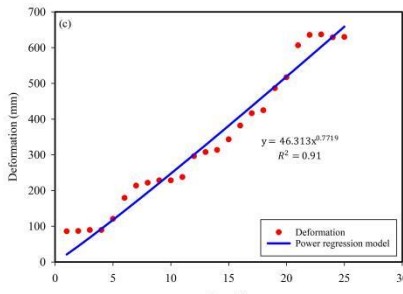

**Figure 6.** The results of regression analysis in the third threshold interval with different regression models. (**a**) Linear regression model; (**b**) exponential regression model; (**c**) power regression model.

The coefficients of determination of the linear regression model, exponential regression model, and power regression model were 0.97, 0.95, and 0.91, respectively. Therefore, the regression model in the third threshold interval was similar to a linear regression model.

Figure 7 shows the results of regression analysis in the third threshold interval with different regression models. The results of the regression relationship are shown in Formulas (19)–(21).

$$y = 0.4196x + 23.194 \tag{19}$$

$$y = 18.154e^{0.0152x} \tag{20}$$

$$y = 3.88476x^{0.6375} \tag{21}$$

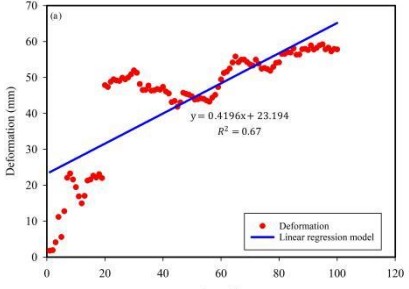 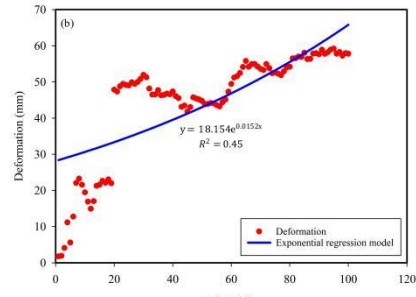 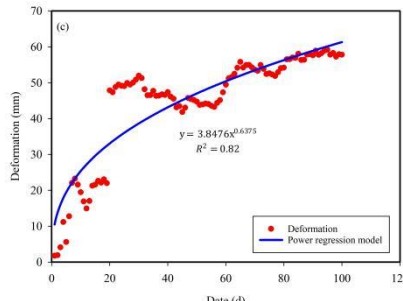

**Figure 7.** The results of regression analysis in the fourth threshold interval with different regression models. (**a**) Linear regression model; (**b**) exponential regression model; (**c**) power regression model.

The coefficients of determination of the linear regression model, exponential regression model, and power regression model were 0.67, 0.45, and 0.82, respectively. Therefore, the regression model in the fourth threshold interval was similar to a power regression model.

Therefore, through the regression model, the thresholds for the early warning levels such as blue warning, yellow warning, orange warning, and red warning on the daily scale and point scale were identified for different warning threshold intervals. The accuracy of the model could be obtained by comparing the identified occurrence times with the actual occurrence times in 2019. The accuracy of the model is shown in Table 4.

**Table 4.** The accuracy of the models with point scale with small scale.

| Warning Levels | Actual Times (d) | Predicted Times (d) | Error | Accuracy |
| --- | --- | --- | --- | --- |
| Blue warning | 8 | 31 | / | / |
| Yellow warning | 110 | 99 | 10% | 90% |
| Orange warning | 2 | 3 | 50% | 50% |
| Red warning | 25 | 43 | 72% | 28% |
| Non-occurrence | 163 | 132 | 19.02% | 80.98% |
| Occurrence | 145 | 176 | 21.38% | 78.62% |

According to the above methods, the regression model was used to evaluate the accuracy of the early warning threshold intervals at different spatial scales with different scales. The results of evaluation accuracy are shown in Table 5.

**Table 5.** The results of evaluation accuracy at different spatial scales with different scales.

| Spatial and Scale | Blue Warning | Yellow Warning | Orange Warning | Red Warning | Occurrence | Non-Occurrence |
| --- | --- | --- | --- | --- | --- | --- |
| Point scale and 100 grids | /– | 90% | 50% | 28% | 78.62% | 80.98% |
| Linear scale and 100 grids | / | 88.89% | 94.74% | 100% | 79.37% | 85.71% |
| Surface scale and 100 grids | 84.21% | 100% | 62.96% | 100% | 92.41% | 93.25% |
| Point scale and 200 grids | 36% | 85.19% | 81.77% | 72.55% | 89.63% | 87.5% |
| Linear scale and 200 grids | 47.89% | 88.89% | 20% | 100% | 91.84% | 84.62% |
| Surface scale and 200 grids | 57.14% | 91.67% | 100% | 100% | 97.44% | 97.22% |

Table 5 shows the accuracy of the early warning threshold model at different spatial scales with different scales. As can be seen from Table 5, when the number of grids in the slope DEM model was 100, the accuracy of predicting non-occurrence in the slope at different spatial scales was 80.98%, 85.71%, and 93.25%, respectively, and the accuracy of the predicting occurrence was 78.62%, 79.37%, and 92.41%, respectively. Therefore, when the number of grids in the slope DEM model was the same, slope deformation on the surface should be used as the most appropriate sliding threshold. When the number of grids in the slope DEM model was 200, the accuracy of predicting non-occurrence in the slope at different spatial scales was 87.5%, 84.62%, and 97.22%, respectively, and the accuracy of predicting occurrence was 89.63%, 91.84%, and 97.44%, respectively. Therefore,

slope deformation on the surface with a large number of grids should be used as the most appropriate threshold when the slope deformation has the same spatial scale but a different number of grids.

### *3.4. Discussion*

The methodology in this study has a number of distinctive characteristics in relation to the early warning threshold model in the field of slope collapse. This method takes speed and acceleration at different spatial scales into consideration. Extensive slope deformation data are systematically collected, which is a scientific and reliable data resource enabling the determination of the early warning thresholds. The slope deformation data used here mainly come from ground-based synthetic aperture radar data. In addition, we considered not only the influence of spatial scales but also the impacts of the scales. Finally, the early warning thresholds at different spatial scales with different scales was explored using the proposed approach, and the accuracy of the model was assessed by using the linear regression model, exponential regression model, and power regression model.

The method proposed by our study adequately considered the influence of different scales and removed the interference of human factors. This is an advantage over existing studies, which have not considered the interference of human factors. For example, S Loew et al. specified the early warning level for the official alarm and evacuation alarm for monitoring slopes, and the early warning thresholds were mainly set by the deformation speed or acceleration [39]. Their method showed that the determination of the early warning thresholds contained factors that could be influenced by humans. Compared with their study, the method presented in our study removed the interference of human factors, and the identification of the early warning thresholds was mainly affected by different scales.

As for the accuracy of the model proposed by our study, the results showed that the early warning thresholds determined by the surface scale on a large scale was the most accurate and should be used as the appropriate early warning thresholds to specify the early warning levels. This is consistent with existing research by S Naidu et al., who established a combined cluster and regression model to identify the early warning thresholds by rainfall thresholds [40]. Their results showed that there was a linear relationship between the early warning thresholds and rainfall thresholds, and they obtained the early warning thresholds at the surface scale through the rainfall thresholds. Compared with their study, the results presented in our study determined the early warning thresholds at different scales, and the early warning thresholds at the surface scale on a large scale were the most accurate. However, their results were mainly restricted to a single spatial scale. In contrast to their results, our results adequately considered the impact of different scales.

In summary, establishing the early warning threshold model as a novel method improved the early warning capability and avoided the occurrence of misjudgment. The results from our proposed model revealed the difference in the early warning thresholds at different scales. The early warning thresholds at the surface scale on a large scale should be used as the appropriate early warning thresholds, which is of great significance to the management, prevention, and emergency rescue of slope collapse.

### 4. Conclusions

Unlike traditional early warning threshold models for slope deformation, the early warning threshold model proposed by our study first introduces the principle of Fisher optimal segmentation, which can successfully identify the early warning threshold at different scales. Our method selected the appropriate sliding threshold as the deformation prediction model and solved the problem of misjudgment of the early warning signs. Using this method to determine the early warning thresholds can reduce the interference of human factors and effectively improve the early warning capability to a great extent. We concluded that the accuracy of the model was influenced by the spatial scale and number of grids, and the accuracy of the early warning thresholds was best when the scale was

at the surface level with 200 grids. The accuracy of the early warning thresholds was worst when the scale was point level with 100 grids. In general, the accuracy of the early warning thresholds at the same spatial level was better when the number of grids was 200, and the accuracy of the early warning thresholds at the same spatial level was worst when the number of grids was 100. Our experimental results revealed that to ensure that the early warning thresholds were the most accurate, the results at the surface scale with 200 grids should be used as the early warning threshold for steep slopes. In the future, more time scales, such as hours and minutes, and more landslide stages, such as rapid stage and instantaneous collapse stage, should be considered to verify the feasibility of the early warning threshold model.

**Author Contributions:** Data analysis, T.L.; formal analysis, X.L., J.W. and D.C.; funding acquisition, J.Q., G.Q. and W.W.; investigation T.L. and B.L.; methodology, X.L., D.C. and L.Z.; validation, T.L., J.W. and X.L.; writing—original draft preparation, X.L.; writing—review and editing, J.L., T.L. and L.Z. All authors have read and agreed to the published version of the manuscript.

**Funding:** The study was financially supported by the Agricultural Science and Technology Innovation Program of the Chinese Academy of Agricultural Sciences.

**Institutional Review Board Statement:** Not applicable.

**Informed Consent Statement:** Informed consent was obtained from all subjects involved in the study.

**Data Availability Statement:** Data available on request due to restrictions eg privacy or ethical. The data presented in this study are available on request from the corresponding author. The data are not publicly available due to the signing of a non-disclosure agreement.

**Acknowledgments:** We acknowledge the reviewers who helped us in the review process.

**Conflicts of Interest:** The authors declare no conflict of interest.

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
