# Peer review of "The Method of Segmenting the Early Warning Thresholds Based on Fisher Optimal Segmentation"

_land, doi:10.3390/land12020344_

Round 1
Reviewer 1 Report (New Reviewer)
The paper is well written in a scientific sense, you just need to correct a few things in my opinion.
I think that you should insert a picture of the whole country of China in picture 2, so that you can see in which part of China the research area is located.
In chapter 2.2. Data source, write which resolution you used for DEM and for orthophoto (line 113 and 114)
Title of table 2 to be on the same page as the table
Author Response
Please see the attachment

Reviewer 2 Report (New Reviewer)
This manuscript attempts to derive an early warning threshold for sliding displacements. However, the overall organization of the whole manuscript was confusing. The authors intended to construct an early warning threshold for a single slide or a regional area. It is different. Which specific type of landslide needs to be presented more explicitly because the deformation and displacement mechanisms differ between the different slide types. All this information is not well presented. The introduction also fails to provide an overview and summary of the existing landslide early warning study efforts. The design of the experiment as a whole is flawed in that some samples are used to propose warning thresholds, and additional independent samples are needed to validate the proposed thresholds.
Author Response
Please see the attachment

Reviewer 3 Report (New Reviewer)
This paper has a clear idea on the whole, and calculates the threshold of high slope landslide warning quantitatively, which provides a reference for disaster prevention and mitigation work.
In my opinion, there are still some problems that need to be improved.
1. It is suggested to modify the overview of the study area. Insert a map of China in Figure 2, so as to see which region of China the study area is located in.
2. The study realized the early warning threshold of high slope landslide at different spatial scales, but how to calculate the accuracy in Figure 5 needs to be introduced in detail in the method section.
3. It is suggested to summarize the status quo of existing early warning research in the introduction.
4. It is necessary to clearly introduce what kind of landslide in specific stage is suitable for research, because the deformation mechanism of landslide in different displacement stages is different, and not all displacement stages are suitable for this threshold.
5、The abstract section:That's a better description“It is necessary to chose”…,not “Overall, we should choose”
6、The aim and the tasks must be highlighted at the end of the Introduction section.
7、The novelty of the paper must be highlighted in the conclusions section.
I wish that my comment would be helpful in improving the quality of this research.
Author Response
Please see the attachment

Reviewer 4 Report (New Reviewer)
I read this manuscript carefully. Indeed, the approach is interesting, but... unfortunately, I find many flaws. I see formulas being used which unfortunately do not have any confirmation through actual engineering geological properties. Landslides are of different types and occur under different conditions, for example, slope dips, particle size of soils, plasticity, hydrogeological conditions… Is this research related to rheology? And what is the practical value of it? Also movement mechanism – rotational, translational, lateral spread, etc. This has been overlooked. Such a generalization of the conditions is simply not correct.
Also some mathematical formulas are used without any coverage with reality. For example, what are these critical thresholds, for example 1038.16 mm? The dimensions and accuracy are somehow disproportionate... What is this super accuracy 0.01 mm in a case we have a displacement of 1 meter…..
I am sorry, but I could not accept this manuscript.
Round 2
Reviewer 2 Report (New Reviewer)
The revised manuscript has been significantly improved.
Author Response
Please see the attachment

This manuscript is a resubmission of an earlier submission. The following is a list of the peer review reports and author responses from that submission.
Round 1
Reviewer 1 Report
The manuscript tries to adopt the Fisher optimal segmentation method to detect the early warning threshold for slope failure based on deformation speed and deformation acceleration. Thought the authors have illustrated the approach through a detailed example, there are still many issues need to be clarified.
1 Though the detection of warning threshold for slope failure is not easy, but it has been dealt with by lot of researchers with different mathematical models. The authors should cite them and tell us why they have chosen the fisher optimal segmentation method in the introduction part.
2 The paper is not well written. The English writing should be improved, especially for section 2. The quality of these figures should be improved.
3 The author verifies their model with a set of data. However we do not know what these data come from? They are slope deformation data for which region/slope? How the authors get these data? Are these data correct?
4 As the Fisher optimal segmentation method is a kind of classic optimization methods, it is not clear for this reviewer that what are the main contributions or innovative parts of this work, .
Reviewer 2 Report
The manuscript is poorly written, and neither is the study not organized well.
The subject is not a national or regional topic. But the references list is full of domestic articles, most of which cannot be found on the internet.
The introduction part does not describe the progress of international studies in the field of warning thresholds. The authors do not review quantitively.
Ch 2. cannot help to understand the methodology of the entire study. Figure 1 is not explained in any words.
